# How Can Curricular Elements Affect the Motivation to Study?

Catherine Bopp [1], Aline Salzmann [1], Silke Ohlmeier [1], Melanie Caspar [1], Erik Schmok [2], Sara Volz-Willems [1], Johannes Jäger [1] and Fabian Dupont [1,*]

1   Department of Family Medicine, Saarland University, 66424 Homburg, Germany
2   Institutional Partnerships, Amboss, 10119 Berlin, Germany
*   Correspondence: fabian.dupont@uks.eu

**Abstract:** (1) Background: This qualitative study aimed to identify and describe course components which affect a student's motivation to learn within a blended-learning competency-based curriculum. (2) Methods: The data were gathered via two consecutive semi-structured group interviews. The participants were purposefully sampled from medical students attending the Family Medicine (FM) class at Saarland University (UdS) in Winter 2020. The two interviews were transcribed verbatim and inductively analysed using content analysis. (3) Results: Three categories of curricular components that affected motivation were inductively formed: (a) the provision of structure (curriculum design), where providing external learning milestones to self-regulated learning positively influenced an interviewee's learning motivation; (b) the provision of interpersonal interactions and emotional relatedness by staff, where constructive feedback and enthusiasm from a teacher facilitated intrinsic motivation and real-life examples helped the students to remember content more easily; and (c) perceived gain in self-efficacy, where a participant's motivation to learn a particular subject area was especially high if it appeared to be highly relevant to practice or exams and the applicability of the knowledge gained was readily apparent. (4) Conclusions: It is important for educators to be aware of how they influence a student's motivation. This study may help to provide an orientation on what to avoid and what to include in a curriculum design project to purposefully foster motivation in students.

**Keywords:** motivation; education; medical; undergraduate; learning; family practice; qualitative research; competency-based; blended-learning; NKLM; curriculum redesign

## 1. Introduction

In the context of sociodemographic shifts and rapid technological advancements, the landscape of practicing medical for future physicians is poised to undergo profound transformations. With the emergence of digital breakthroughs and artificial intelligence, the challenges faced by the next generations of physicians in family medicine will be markedly distinct from those encountered in the past. Specifically, digital diagnostic tools hold the potential to assume a substantial portion of medical tasks, thereby redefining the scopes and responsibilities of healthcare professionals [1]. In recent years, the idea of competency-based medical education (CBME) has been embraced by organizations around the world in the urgent need to adapt teaching in medicine to this new work environment and to prepare students adequately for high competency levels in the workplace [2]. However, the development of CBME has been hampered by slowness and a variety of implementation issues that differ greatly between countries and institutions. Unanticipated changes to medical education occurred during the COVID-19 pandemic all over the world. At Saarland University, this period helped with the implementation of new curricular elements based on CBME. To this day, researchers believe that the basic principles of CBME need to be further developed [3,4].

In Germany, faculty members are mandated to facilitate the implementation and enhancement of the national CBME catalogue of learning objectives (NKLM) (current version: NKLM 3.0 (in press)) [4].

In the process of devising and investigating Saarland University's (UdS) novel CBME curriculum in family medicine (FM) in 2020, motivation emerged as a pivotal psychological construct. The self-determination theory (SDT) distinguishes three types of motivation: IM, extrinsic motivation (EM), and amotivation [5]. As defined by Deci, E.L. (1973), in contrast to EM, IM is not driven by external rewards and is associated with improved learning outcomes and performance [5,6]. Hence, the substantial influence of intrinsic motivation (IM) on the fundamental objectives of enhancing teaching quality through the NKLM educational redesign strategy became apparent. The significance of motivation in CBME was further reinforced by relevant review papers [7]. Particularly noteworthy are the essential prerequisites that must be fulfilled for the emergence of intrinsic motivation as they play crucial roles in assessing student needs. According to self-determination theory, the fulfilment of three fundamental psychological needs—competence, autonomy, and relatedness—constitutes the primary prerequisite [5]. While instructional strategies for fostering autonomy and promoting the development of intrinsic motivation have been refined based on self-determination theory [8], a comparable guide for addressing competence and relatedness remains absent to date.

A significant challenge that arose during the implementation of the novel CBME curriculum was the need to navigate strict contact restrictions and enforce measures to safeguard the well-being of both students and simulated patients amidst the backdrop of the COVID-19 pandemic. Another challenge was that despite the NKLM's explicit catalogue of competencies, to date, it has not yet been clarified which instructional design might be the most suitable to meet its own requirements. In its latest publication of the NKLM, the Medical Faculty Association of the Federal Republic of Germany explicitly called upon the faculties to actively contribute to the development of instructional design. The publication emphasised the need for the careful evaluation and assessment of new events and examination formats to effectively incorporate and anchor the specified competencies within the curriculum [9].

This study can be seen as a first step towards this call for action. It aimed to find out to what extend the individual elements of the instructional design are useful (or not) for the enhancement of IM in a CBME undergraduate family medicine curriculum based on SDT. To promote perceived competency, students were expected to manage hybrid chief-complaint-based patient simulations, during simulation based seminars. Patient-cases were standardised by using professional patient actors and 72 self-written cases. This approach aimed to foster practical training, whole-task learning, and near-workplace-based learning, thereby enhancing students' overall competence. [10,11]. The second driver of IM is autonomy. For that purpose, we used a blended learning concept where on-site learning was supplemented by self-regulated online learning stages [12]. We provided a centralised learning platform (a newly created landing page), which included articles, embedded lectures, commentary on guidelines, podcasts, and multiple-choice question sessions, in a modular setup (e.g., abdominal pain, febrile infections, and back pain). The goal was to allow students to choose and proceed at their own rate when preparing for on-site sessions [13]. The third driver of IM is relatedness. To help students connect with the profession, the aforementioned simulations provided them with realistic family medicine daily life tasks that they were required to execute independently while obtaining professional feedback from real-life family medicine specialist-physicians and instructors. For more information regarding the construction and defined learning outcomes of the new FM curriculum at UdS, please see Figure 1 or refer to the study by Salzman et al. (2022) [14].

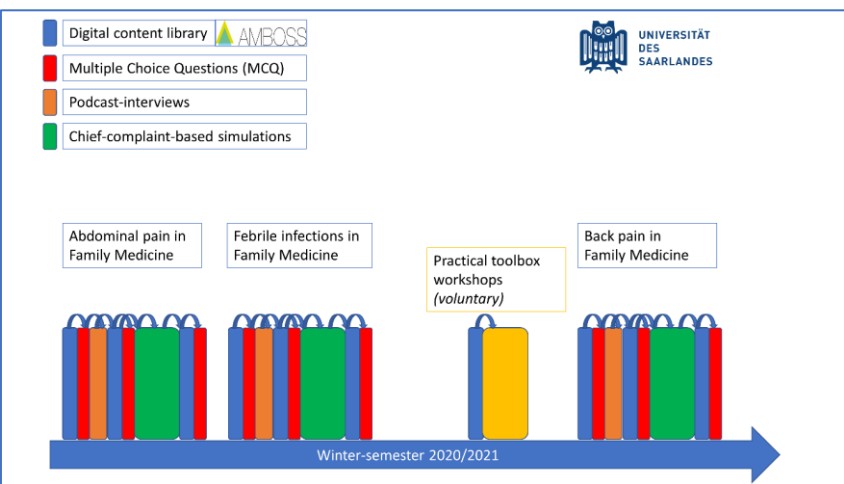

**Figure 1.** Modular (chief-complaint-based) setup of our year five CBME blended-learning family medicine curriculum at Saarland University, Germany.

Another important element that defines how students approach learning tasks is their perceived self-efficacy [15]. Students with a strong sense of self-efficacy appear to have a high belief in their own capabilities. They perceive tough assignments as challenges, and they may be more intrinsically interested in complex learning activities than students with lower levels of perceived self-efficacy [15]. The premise that students approach learning differently, even while part of the same curriculum, is referred to as "approaches to learning" (ATLs) in current medical education research [16–19]. Based on Marton et al.'s work, three different ATLs could be identified: in the deep approach, the learner is intrinsically motivated to achieve a deep understanding of the field of study [19]; the strategic approach is characterised by organised study with a focus on achievement [19]; and the surface approach is associated with EM, a lack of purpose, and primarily memorizing content, often superficially [19]. An ATL is not seen as a stable trait but as a dynamic feature which is influenced, for example, by individual experiences [19].

While learning approaches are also seen as individual and variable, motivation can often be influenced more directly [5]. The extent to which specific elements of the new curriculum contributed to or hindered the development of student motivation remains uncertain. This study endeavoured to address this knowledge gap by qualitatively exploring and delineating the various instructional design elements and their impacts on a student's motivation to learn. It was part of the change progress towards the implementation of the new NKLM and CBME in undergraduate FM in Germany.

## 2. Materials and Methods

### 2.1. Study Design

This study applied a constructivist qualitative method to better understand the internalizing and externalizing factors affecting motivation in various curriculum elements, taking into account students' ATLs [19,20]. Data were collected during two consecutive semi-structured group interviews [21]. The conversations between the participants were directed to their motivation to study within the new curriculum in FM.

### 2.2. Participants and Setting

The curriculum was introduced as a curriculum redesign project in the winter semester of 2020–2021 at the Department of FM at UdS, Germany. Prospective participants were selected out of 92 enrolled year 5 students and invited via email. Maximum variation sampling, based on the quantitative questionnaire results from a quantitative study (including the Revised Approaches of Studying Inventory (RASI) and age), was used to guarantee a higher level of heterogeneity of the participants and their study behaviours [18,19,22]. The RASI consists of 52 items in total and is part of the overall ASISST questionnaire

(Approaches and Study Skills Inventory for Students). For improved usability, three items were chosen from each of the three scales (approaches, shown in Table 1), with scores ranging from 3 to 21, based on a 7-point Likert scale (1 = does not correspond at all and 7 = corresponds exactly) [19]. The items used and their Cronbach's alpha values can be seen in Table 1 [19,23]. Confirmations of the validity of similar approaches were completed in previous publications [24]. From the 92 enrolled students in the course, 84 completed at least one of the RASI item-questions. A simplified allocation strategy (i.e., an alternative to cluster analysis) was used to identify the students with high expressions of one ATL out of these 84 students. Participants were allocated to one ATL if they reached at least 60% of the maximum score in one ATL and had at least a 2-point difference in all scores in the remaining two categories [23]. Based on random sampling, ten students for each of the three ATLs were invited, as were four participants from the category under 25 years and four participants from the category over 25 years. Qualitative group interview recruitment was stopped when, for each group interview, two students from each ATL and one student above and one under 25 years agreed to participate ($n$ = 16). A flowchart of the sampling strategy is displayed in the study by Bopp et al. (2022) [23].

**Table 1.** The items used, their original item number in the RASI survey, and their Cronbach's $\alpha$ value (by approach).

| Approach | Item | Subscale | Item Number RASI | Cronbach's $\alpha$ |
|---|---|---|---|---|
| Deep approach | It is important for me to be able to follow the argument or to see the reasons behind things. | Use of evidence | 49 | 0.618 |
|  | I sometimes get 'hooked' on academic topics and feel I would like to keep on studying them. | Interest in ideas | 52 |  |
|  | Before tackling a problem or assignment, I first try to work out what lies behind it. | Seeking meaning | 43 |  |
| Strategic approach | I work steadily through the term or semester rather than leave it all until the last minute. | Time management | 31 | 0.548 |
|  | It is important to me to feel that I am doing as well as I really can in the courses here. | Achieving | 10 |  |
|  | I keep an eye open for what lecturers seem to think is important and concentrate on that. | Alertness to assessment demands | 41 |  |
| Surface approach | Often, I find myself wondering whether the work I am doing here is really worthwhile. | Lack of purpose | 3 | 0.616 |
|  | I am not really sure what is important in lectures, and so I try to get down all I can. | Unrelated memorising | 32 |  |
|  | Often, I feel I am drowning in the sheer amount of material we are having to cope with. | Fear of failure | 8 |  |

The interviews were conducted in the German language and took place in the form of online Zoom conferences on 18 December 2020 and 7 January 2021. The average duration of the interviews was 110 min (118 min and 102 min), including introductions to those present and a brief explanation of the process. In total, out of the 16 selected interview partners (students), 2 did not take part in the interviews. In the first interview, one of the eight participants (under 25 years) missed the event. This resulted in a final number of participants of $n$ = 14. In the second interview, only one participant from the surface approach took part. The participants agreed to participate during the survey, and again, they verbally agreed to the videorecording, transcription, and storage of the discussion for data analysis and publication. The interviews were moderated by C.B. and A.S., who were both doctoral students at the Department of FM, UdS.

*2.3. Data Collection*

The interviews were led based on a semi-structured interview guide. Follow-up questions were asked to stimulate further explanation. By consensus of the three main coders (C.B., A.S., and F.D.), theoretical saturation was reached after the second interview. Based on the study by Saunders et al., no further aspects related to the research question could be extracted from the data except for those that could be assigned to the three main categories, and the same codes appeared again [25]. The interviews were audio- and video-recorded and transcribed verbatim. The identities of the interviewees were pseudonymised in the transcripts. Member-checking with volunteers from the interviewees was completed at the end of the interviews after a 3 min summary from one of the moderators and again with 2 interviewees after transcription and coding. No amendments were made by the participants.

*2.4. Data Analysis*

Qualitative content analysis was applied for the analysis of the transcripts [19]. C.B. inductively analysed the two interviews using a line-by-line, open-coding strategy with the software MAXQDA 2020. Focused coding was completed by C.B. and two other researchers (A.S. and F.D.) during investigator triangulation sessions. Afterwards, the preformed codes were revaluated under the consideration of the theoretical framework of motivation in two interdisciplinary qualitative research sessions at the institute of FM (UdS) with S.V.-W., M.C., S.O., C.B., A.S., and F.D.

## 3. Results

*3.1. Demographics*

Five of the interviewees were male and nine were female. Three were above the age of 25 and eleven participants were under the age of 25. The shortened RASI identified four participants as deep-approach-learners, four as strategic-approach-learners, and three as surface-approach-learners, and two could not be clearly allocated to one specific ATL. In interview one, there was one international student from the ERASMUS program. The other students were all normally enrolled full-time students.

*3.2. Identified Factors That Altered a Student's Motivation to Learn*

For each of the two interviews, an inductive code system was established separately, and the two systems were then compared and put together. The combined code system was structured into three main categories (curricular units) and several subcodes (more concrete elements mentioned by the students as motivating/positive or demotivating/negative) (Table 2).

3.2.1. Provision of Structure by Curriculum Design

A curriculum's scaffolding (structure) has been acknowledged as a powerful motivator. The participants were particularly impressed by the clearly defined educational objectives and clearly communicated, interspersed short-term targets (Table 3: interview one, transcript position 31). Although the students appreciated the blended learning approach's flexibility in terms of time and location (Table 3: interview two, transcript position 68), the external rhythms provided by the curriculum, such as fixed appointments, appeared to be important to some participants in order to motivate them to learn (Table 3: interview two, transcript position 66)). In contrast, the motivation to study was diminished when the participants experienced cognitive overload, such as when the course content was deemed too lengthy (Table 3: interview one, transcript position 95). Detailed results concerning course structure are published separately [23].

**Table 2.** Identified aspects of a course that were linked to motivation (the selection of created codes in MAXQDA 2020).

| Categories | Provision of Structure by Curriculum Design | Provision of Social Interactions and Emotional Relatedness by Staff | Gain in Perceived Self-Efficacy |
|---|---|---|---|
| Positive effect on motivation (+) * | + accessibility<br>+ clear course structure<br>+ thorough instructions<br>+ use of different media (audio, video, etc.)<br>+ flexibility and autonomy<br>+ fixed appointments for external rhythm | + real-life case vignettes<br>+ enthusiastic teacher<br><br>+ reasonable feedback<br>+ opportunity to ask questions<br>+ being invited to reflect on a subject | + gain practical experience<br>+ feel familiar with basic competencies<br><br>+ applicability and relevance to everyday clinical practice is evident |
| Negative effect on motivation (−) * | − content overload | − abstract content with no examples provided<br>− purely negative feedback<br>− fear of failure in front of others | − applicability and relevance in clinical routine are not evident |

* The subcodes could be assigned to multiple categories in MAXQDA, and the overlap has been marked by + or −.

**Table 3.** Statements of the interviewees quoted in the text (translated from German by C.B.).

| Provision of Structure by Curriculum Design | Provision of Social Interactions and Emotional Relatedness by Staff | Gain in Perceived Self-Efficacy |
|---|---|---|
| B2: '(…) you get it [the topics] as bite-sized chunks, that's what I found very convenient. (…) The other point I liked was, that it was structured (…)' (interview one, transcript position 31) | B13: 'I liked that the podcasts weren't a live event (…) I prefer to manage my own time. (…) But if it would be more interactive (…), it might work better as live event in order to interact and answer to questions, whether it is a video-chat or a live lecture, I don't really care.' (interview two, transcript position 68) | B1: '(…) if I master the things I am learning in this course, I will feel more competent when friends ask me: "I have this or that [medical problem]. What should I do?" because these are the disease patterns, a family practitioner sees often.' (interview one, transcript position 47) |
| B8: 'I always study at the last minute and therefore I prefer scheduled events.' (interview two, transcript position 66) | B10: 'It's easy for me to listen to lectures and motivate myself (…) when I can see the teacher and his facial expressions and feel his presence. (…) I miss the human component if there are just slides and sound [in an online lecture].' (interview two, transcript position 69) | B1: 'every doctor should be able to master the basics. And it [the course] was focussed to that and I liked that.' (interview one, transcript position 29) |
| B3: 'I found this [the seminar preparation materials] quite time consuming after all (…). And for me, this led to an uninterested attitude towards the simulation session because I had the feeling that you were expecting too much!' (interview one, transcript position 95) | B2: 'So you revise the given content carefully because you don't want to be embarrassed there [(in the simulation session].' (interview one, transcript position 132) | B8: 'Anamnesis and physical examination, I can definitely do myself, but for the diagnosis I would rather get feedback if it is correct (...). In a real environment, it's good to have someone looking over your shoulder (…)' (interview two, transcript position 170)<br>B12: '(…) our performance wasn't even that bad, but the feedback was purely negative. That's what I found a bit annoying (…)' (interview two, transcript position 222)<br>B1: '(…) that you don't enter into a subject from the side of the illness (...). How does a patient get to the hospital? (...) He doesn't say: 'I have pneumonia (...)' (...) but 'I can't breathe' or 'I have a fever'. (...) that is much more practice-oriented.' (interview one, transcript position 45) |

### 3.2.2. Provision of Social Interactions and Emotional Relatedness by Staff

As one participant indicated, discussions with a teacher, such as responding to a teacher's questions, is a clear advantage for the learning process (Table 3: interview two, transcript position 68). Another participant stated that the nonverbal parts of communication were also crucial for her motivation (Table 3: interview two, transcript position 69).

The main reason students wanted live events in addition to self-regulated online learning phases appeared to be social interaction (Table 3: interview two, transcript position 68).

In contrast to the favourable impacts of social interaction, the presence of peers and professors caused some students to feel slightly nervous while participating in simulations taking place on-site. On the other hand, participants were more eager to prepare and learn before the simulations to avoid embarrassing situations (Table 3: interview one, transcript position 132). Furthermore, there appeared to be a transfer of motivation from teaching staff to students. One interviewee said: '(. . .) we had this professor, and she was so passionate about her subject (. . .) this is what is missing on the [online] learning cards, the enthusiasm for studying, for the subject and why one is doing it.' (interview two, transcript position 121). Another interviewee added: '(. . .) when a teacher is highly motivated to teach his subject, he conveys this enthusiasm to the students.' (interview two, transcript position 121).

The participants said they found it easier to access learning content when it was well-presented and had clear links to reality. One participant (ATL: deep approach) explained her need for concrete content: '(. . .) I always need an example that I can remember. They [real life scenarios] make the subject more impactful (. . .).' (interview two, transcript position 51).

### 3.2.3. Gain in Perceived Self-Efficacy

The participants from the deep-approach type explained that they were motivated to learn the FM content because it made them feel more confident to deal with ordinary day-to-day medical enquiries (Table 3: interview one, transcript positions 47 and 29).

Frequently, when medical students are engaged in performing a task they feel uncertain about, they perceive the presence of a supervisor as beneficial (Table 3, interview two, transcript position 170). When receiving positive feedback, students seemed to experience growth in perceived self-efficacy, and one interviewee reported the following from their simulation seminar: 'Especially the feedback, (. . .) that we performed better than we anticipated. This was a really good experience.' (interview one, transcript position 199).

One participant responded defensively upon receiving solely negative feedback. Despite not seeking to rationalise the evaluation of her performance, there was a noticeable decline in her belief in her own capabilities (perceived self-efficacy). This observation suggested that an exclusive focus on negative feedback may impede the development of perceived self-efficacy, consequently affecting intrinsic motivation (Table 3: interview two, transcript position 222).

When students perceived the possibilities for knowledge application in real-life settings, they appeared to accept the need to study them more readily. Participants in both interviews emphasised "that it [the FM course] is focused on practice (...)" (Table 3: interview one, transcript position 45). In contrast, if the students believed that the subject was unimportant to their future careers, their enthusiasm for studying declined. One student said, "(...) things you never heard of and will probably never hear of again, (...)", and another commented the following about the lecture notes from another subject, "(...) nobody could possibly cram that!" (interview one, transcript position 39).

### 4. Discussion

A structured curriculum that leaves students mostly free to choose when and where to study but also provides external rhythm and structure, interpersonal interactions, and emotional relatedness by staff appears to be capable of fostering a student's IM within a CBME curriculum. A blended learning curriculum, which combines online learning and face-to face learning, appeared to meet these requirements in our setting.

As far as the choice of learning activities for independent learning was concerned, the choice of learning activities itself seemed to play less of a role in motivation than the content and its visual appeal. While it was seen as positive that there was a certain amount of variety (podcasts, video lectures, and learning cards/commentary), it appeared to have

had a negative effect on self-efficacy when the materials provided were seen as too time-consuming to process. Teachers planning the implementation of a CBME curriculum and seeking to maximise learning activity diversity to appeal to as many students as possible should be aware of this possible negative effect.

Concerning the choice of content, Ryan et al. believe that it is futile to expand broad-based training [3]. Learners may not benefit because much of the content would not be directly relevant to their daily practice.

Our research showed that focusing on topics that students think are meaningful for their future professional life appeared to be helpful in stimulating their IM and increasing their perceived self-efficacy. We agreed with Ryan et al. that teachers should try not to overload students with excessive content [3]. However, we found that it appeared to be possible to emphasise clinical relevance through simple methods, such as linking topics to relevant clinical cases, providing commentary, or enabling interactions with real life specialist physicians in chief-complaint-based sessions.

COVID-19 made practical learning using real life patients almost impossible. While adhering to the imposed hygiene measures, it was possible to conduct simulation seminars with professional patient actors. It turned out that simulation seminars appeared to be a good fit for CBME. Students in our interviews were more motivated if they experienced a positive development in their perceived self-efficacy during such simulation seminars (Table 3). In particular, the opposing effects of providing good (meaningful) vs. bad (purely negative) feedback at the end of the simulations appeared to play a major role. This finding confirmed one of the postulations made by Kursukar et al. (2011) [16], here they found that negative feedback lacking in constructive components can make students lose faith in their capabilities and demotivate (loss in self-efficacy) [16].

In addition, the seminars appeared to be effective in teaching important psychomotor skills as well as affective skills, such as interpersonal interaction and empathy. These skills, which are often taught only as part of the hidden curriculum, seemed to have an additional motivational aspect for students.

In our interviews, the students indicated that affective skill-learning also occurred during lectures (Table 3: interview two, transcript position 69 and interview two, transcript position 121). These emotions appeared to help students feel connected to the subject and increased their relatedness, which fostered their IM, as postulated by SDT [5]. It remained unclear whether there was a relevant difference between the on-site lectures and the online or on-demand lectures in terms of how effective they were in motivating the students. Future research may be needed, perhaps during a period with no hygiene restrictions.

The desire for supervision and interaction between a student and their teacher expressed in the interviews showed the important role of the teacher. It appeared that the teachers conveyed enthusiasm but also served as role models in helping students achieve the intended learning goals.

*Limitations*

Even though theoretic saturation was reached, this qualitative study provided statements from only 14 individual students that could not be seen as universally applicable. The aim of this study was to better understand what may help students in an undergraduate CBME FM curriculum, especially for those implementing the new NKLM standards. We did not aim to objectively measure how much intrinsic motivation has changed in students, but rather, we sought to qualitatively discover which components in the new FM curriculum at UdS were particularly important for this change. For this reason, this study deliberately examined the students' subjective opinions [21]. To increase credibility, future studies may want to extend the study cohort to a larger population of students, possibly at different medical schools with comparable programs.

The new CBME curriculum in FM described in this study was introduced in the middle of the COVID-19 pandemic. This pandemic has had a significant impact on higher education. Face-to-face classes were replaced with online learning as a result of campuses

being abruptly closed as a social distancing measure. This may have shaped students' views of both on-site and online learning. The needs of students may have differed from those during non-pandemic times. In our interviews, it was positively emphasised that there was still face-to-face teaching in the family medicine course, albeit under strict hygiene conditions, whereas most departments at that time had fully transitioned to fully online learning. Under these circumstances, students' emphasis on social learning and the role of the teacher may have naturally grown. The data from this study suggest that the social component plays an important role during the academic learning process in CBME.

The participants' readiness to participate demonstrated clear differences in availability between the different ATL cohorts. Other publications have shown that the deep-approach to learning correlates with IM [26], which might explain why more of the invited students from the deep-approach learning type agreed to participate in interviews. The sampling strategy chosen nevertheless attempted to provide a balanced representation of the different ATL groups. However, further studies should try and lower the threshold to participate in qualitative interviews further to include more diverse participants.

## 5. Conclusions

This study attempted to provide insights into students' inner motivational lives. The results may not be universally applicable, but they may be useful to educators who want to consciously stimulate students' motivation to learn and need suggestions as to what might be possible influencing factors when implementing CBME and NKLM in FM. This study revealed three important aspects that ought to be considered if educators aim to purposefully foster student motivation. These are: The role of instructional design in motivation, the role of emotional motivation transfer between staff and students, and the role of perceived professional growth while learning.

**Author Contributions:** Course implementation, C.B., E.S., A.S., S.V.-W., F.D. and J.J.; conceptualization, C.B., E.S. and F.D.; methodology, C.B. and F.D.; software, C.B.; validation, C.B., M.C. and F.D.; formal analysis, C.B., A.S., M.C. and F.D.; writing—original draft preparation, C.B. and F.D.; writing—review and editing, C.B., A.S., S.V.-W., M.C., S.O., F.D. and J.J.; visualization, C.B.; supervision, J.J. All authors have read and agreed to the published version of the manuscript.

**Funding:** The Department of Family Medicine holds active educational and research cooperation with IMPP (a national government agency for medical and pharmaceutical state exam questions), AMBOSS (a digital editing house), and the Association of Statutory Health Insurance Physicians (a regional body in Saarland of public health insurance physicians). There were no influences on the study's question design or analysis. C.B. received funding from the university graduate international program (GRADUS). A.S. received funding from the national scholarship foundation (Studienstiftung des deutschen Volkes) for their participation at the WONCA world conference in 2021 (digital).

**Institutional Review Board Statement:** Ethical approval was granted by Ethikkommission der Ärztekammer des Saarlandes on 25 September 2020 (BU234/20).

**Informed Consent Statement:** Informed consent was obtained from all subjects involved in the study.

**Data Availability Statement:** The data provided in this work are available from the corresponding author upon reasonable request.

**Conflicts of Interest:** The Department of Family Medicine holds active educational and research cooperatives with IMPP (a national government agency for medical and pharmaceutical state exam questions), AMBOSS (a digital editing house), and the Association of Statutory Health Insurance Physicians (a regional body in Saarland of public health insurance physicians). There were no influences on the study's question design or analysis. One of the co-authors (Erik Schmok) was actively employed by Amboss. He did not have access to the research materials, the data analysis, or the creation of the manuscript. However, he was closely involved in creating this curriculum and enabling the data collection and the curriculum creation and management (e.g., setting up the digital platforms for data collection, etc.).

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
