# Peer review of "How Can Curricular Elements Affect the Motivation to Study?"

_ime, doi:10.3390/ime2030015_

Round 1

Reviewer 1 Report

This is a well written and high quality manuscript relating to the elements of a curriculum for family medicine exploring factors that influence students in their learning.

I have some minor comments that are mainly clarifications, rather than major or structural amendments to the manuscript. 

Firstly, noting the timeframes of the research I cannot see any reference to the COVID-19 pandemic. COVID-19 had a significant impact on the delivery of medical education, particularly in terms of remote learning, and potential change or loss of traditional forms of social learning. I note the authors refer to the importance of role models and inspirational teachers in terms of motivation. Please. Can the authors clarify any impact relating to Covid-19 on their project and data collection, in addition to whether this might have retrospectively shaped participants' views on the relative value of relational learning.

Secondly, the authors describe an intention to create intrinsic motivation. From my understanding, intrinsic motivation is to some degree innate within learners and varies based on an individual's character and deeply seated worldviews and associated personal aspirations. As such there is some question as to whether attitudinal changes can be taught - at least using traditional knowledge/skills based curriculum focussing on a competency based curriculum (and indeed how attitudinal changes if achieved can be measured objectively). I would be grateful for clarification on the authors’ definition of intrinsic motivation and whether the term ‘created’ was intended or developed and enhanced would be more appropriate.

Please state in the methods that the interviews were conducted in German. Please also describe the duration of interviews (range and average). Do the authors feel that data saturation was reached, given the range of themes? I am a little confused about the final N value given attrition - the methods and results differ.  Please justify any decisions on saturation with a reference. A potential publication to help with this may be - Saunders, B., Sim, J., Kingstone, T., Baker, S., Waterfield, J., Bartlam, B., ... & Jinks, C. (2018). Saturation in qualitative research: exploring its conceptualization and operationalization. Quality & quantity, 52, 1893-1907.

The methods describe sampling stopping. I'm unsure whether this means purposive sampling was discontinued, but wider sampling continued, or whether this refers to recruitment stopping. Please clarify

Can the authors provide any further demographic details of the study participants beyond their broad age category?  For instance, are their details of their prior grades leading up to year 5? Are there also any details on the participants ethnicity or status as a German national student or international student at the university? To clarify, was the maximum variation sampling approach according to the participants' learning approach alone or a demographic feature?

As mentioned these are relatively minor clarifications. I feel this is a high quality, well-written piece of work. If the above factors are addressed, I have no further concerns about it being accepted for publication.

Author Response

Dear esteemed reviewers,

We express our profound gratitude for your diligent and insightful assessment of our manuscript. Your valuable recommendations have greatly enhanced the rigor and coherence of our work. On behalf of the entire team, I extend my heartfelt appreciation for your dedicated effort in reviewing our paper. Textual amendments based on your constructive feedback are highlighted in yellow throughout the manuscript, signifying the incorporation of your invaluable suggestions.

Below, we provide a comprehensive response to each of your comments and suggestions:

Reviewer 1; Comments to the Author:

This is a well written and high quality manuscript relating to the elements of a curriculum for family medicine exploring factors that influence students in their learning.

I have some minor comments that are mainly clarifications, rather than major or structural amendments to the manuscript. 

Firstly, noting the timeframes of the research I cannot see any reference to the COVID-19 pandemic. COVID-19 had a significant impact on the delivery of medical education, particularly in terms of remote learning, and potential change or loss of traditional forms of social learning. I note the authors refer to the importance of role models and inspirational teachers in terms of motivation. Please. Can the authors clarify any impact relating to Covid-19 on their project and data collection, in addition to whether this might have retrospectively shaped participants' views on the relative value of relational learning.

We thank you very much for your constructive and encouraging feedback. This is an important remark. Of course, Covid-19 not only had an impact on our curriculum design, but also on student learning. Certainly, the needs of students during this very hard period may have also been somewhat different from non-pandemic times. In our interviews, participants particularly positively emphasized that there was still face-to-face learning in our curriculum. (Of course with stricht hygiene measures, whereas most departments at that time relied completely on online teaching.) The individual perception of social (group) learning might have been increased under these circumstances. We think that this element, even though, maybe expressed more strongly, may still play an important time in post-covid times.

We made sure that this element of bias was addressed adequately in the discussion session.

Secondly, the authors describe an intention to create intrinsic motivation. From my understanding, intrinsic motivation is to some degree innate within learners and varies based on an individual's character and deeply seated worldviews and associated personal aspirations. As such there is some question as to whether attitudinal changes can be taught - at least using traditional knowledge/skills based curriculum focussing on a competency based curriculum (and indeed how attitudinal changes if achieved can be measured objectively). I would be grateful for clarification on the authors’ definition of intrinsic motivation and whether the term ‘created’ was intended or developed and enhanced would be more appropriate.

Your comment is very understandable, thank you. In fact, people are never completely intrinsically motivated or controlled motivated or amotivated. Literature suggests to see it as a fluid state on a continuum scale. Students or participants in general appear to show different levels of expressivity in each of the categorie. Of course, the expressivity within one category may be influenced only to a certain extent. Therefore, the terms "enhanced" or "developed" would be more appropriate than "create." We adapted our wording throughout the text. Some students are highly amotivated and have very little, if any, intrinsic motivation for certain subjects and are hard to motivate. However, it has been shown that this is true for very few individuals in the cohort of medical students in question (Salzmann, Aline, et al. "Can a Family Medicine Curriculum Increase the Attraction of Family Medicine as a Career Choice?" Zeitschrift für Allgemeinmedizin 98.6 (2022): 229-233.) (less than 5%). One way to measure this objectively is, for example, the AMS (Academic Motivation Scale) questionnaire at the beginning and end of the course. Our findings from that approach can be found in (Salzmann et al.). In this study the aim was not to objectively measure how much intrinsic motivation has changed, but rather to obtain indicators of which factors are particularly powerful in changing motivation. For this reason, this study deliberately examines students' subjective (qualitative) opinions. We enhanced the paragraph in the introduction, which describes our previous work.

Please state in the methods that the interviews were conducted in German. Please also describe the duration of interviews (range and average). Do the authors feel that data saturation was reached, given the range of themes? I am a little confused about the final N value given attrition - the methods and results differ.  Please justify any decisions on saturation with a reference. A potential publication to help with this may be - Saunders, B., Sim, J., Kingstone, T., Baker, S., Waterfield, J., Bartlam, B., ... & Jinks, C. (2018). Saturation in qualitative research: exploring its conceptualization and operationalization. Quality & quantity, 52, 1893-1907.

Thank you very much for this comment. We fully understand your confusion. We have gladly included the additional information in the methods section.. We are sorry for this misunderstanding of the number of participants. N=16 refers to the recruited students who had agreed to participate. Since two of them did not show up (no show) in the final interviews, as explained afterwards, this results in a final number of participants of N=14. This final N value was supplemented.

Many thanks for this reference:

Concerning data saturation:

By consensus of the three main coders (CB, AS, FD) it was found that in the second interview, no further aspects concerning the research question could be extracted from the data besides the previously emerged categories. (cf ‘(non)emergence of new codes or themes has been taken by others’ (Birks and Mills 2015; Olshansky 2015 in Saunders et al.2018).

Also, we found in this second interview group that the main themes returned and many of the codes from the first interview were used again (cf. ‘informational redundancy’ (Francis et al. 2010; Guest et al. 2006 in Saunders et al. 2018)).

I hope this commentary helped to clarify your concerns.

The methods describe sampling stopping. I'm unsure whether this means purposive sampling was discontinued, but wider sampling continued, or whether this refers to recruitment stopping. Please clarify.

Thank you for pointing this out. Indeed, only the recruitment was stopped, when the intented sampling size was reached, not the purposive sampling itself. This was corrected in the text. Apologies for the miswording.

Can the authors provide any further demographic details of the study participants beyond their broad age category?  For instance, are their details of their prior grades leading up to year 5? Are there also any details on the participants ethnicity or status as a German national student or international student at the university? To clarify, was the maximum variation sampling approach according to the participants' learning approach alone or a demographic feature?

Data on previous grades are potentially available as they are part of Salzmann et al's collected data. This data was not used for the interview sampling. There was also no correlation between motivation and grades in the quantitative study by Salzmann et al. (see above).

From the participant introduction at the beginning of the interviews, we can see that one student is an Erasmus student at Saarland University. All other participants are normally enrolled German national students. 6 participants  were selected based on their learning approach and 2 participants were selected based on their age category. This was done to adhere to a maximum variation sampling strategy. Interestingly, all of the over-25s had previously completed vocational training.

As mentioned these are relatively minor clarifications. I feel this is a high quality, well-written piece of work. If the above factors are addressed, I have no further concerns about it being accepted for publication.

We thank you very much for your acknowledging words. We hope that we have been able to implement your feedback in a way that is satisfactory to you. We very much look forward to publishing our findings and to help develop medical education further.

[final notes]

Dear reviewers,

We would like to express our sincere gratitude for your valuable feedback and support in reviewing our paper. Your constructive comments and suggestions have significantly enhanced the quality and clarity of our work. We appreciate the time and effort you have invested in providing such insightful reviews.

We have carefully considered all your comments and made the necessary revisions to address the issues raised. In response to Reviewer 1's question regarding the impact of the COVID-19 pandemic, we have added a detailed explanation of how the pandemic influenced our curriculum design and data collection, and how it may have shaped participants' views on relational learning.

Additionally, we have rephrased certain sections to better convey our research intentions and contributions. We have integrated relevant review papers and updated our bibliography to reflect the most recent and pertinent literature in the field of medical education.

In the Methods section, we have provided a clearer explanation of the ASSIST item questions and their connection to the RASI instrument to avoid confusion for readers.

Moreover, we have expanded the discussion section to address the limitations of our study and the implications for practice and research. We have also added a more comprehensive conclusion that summarizes the main findings and their significance in the context of current research.

Regarding the term "blended learning," we have included a well-defined explanation with proper references to ensure a clear understanding of its meaning in our study.

We have also taken the opportunity to clarify points raised by Reviewer 3, including providing additional context for certain statements and refining Table 3 for better comprehension.

Once again, we extend our heartfelt appreciation for your academic input and support of our medical education research. Your contributions have been instrumental in refining our work, and we hope that the improved manuscript now meets the standards for publication in your esteemed journal.

Thank you for your valuable guidance, and we look forward to the possibility of sharing our findings with the wider medical education community.

Sincerely

On behalf of the entire authors team

Reviewer 2 Report

The novelty and contribution of this paper cannot be found inside the text; thus, some changes are needed. The research approach is very weak, and the findings need to be strengthened. Here are my comments on improving the manuscript:

1. Overall:

a)       Why do the authors conduct this study? The research contributions are weak. Please kindly explain.

b)      The research structure is not appropriate for a scientific article, e.g., research questions are missing from the introduction, results and conclusion as well as the discussion is not written in an “appropriate” manner. Please update.

c)       Please consider how to effectively integrate some review papers and update.

2. Introduction:

a)       Research questions, that drive the paper, should be built in the introduction from an ongoing and pertinent bibliography (up to 2022-23) and these should be of global interest and not focused on a particular local problem. Identifying a research gap is the most important by indicating in-text some newer references that are significant to your particular field of research.

b)      Due to a weak contribution, please pay attention to addressing new research gaps and then emphasize why the authors do this study.

3. Discussion:

a)       The proposed reason for conducting such a study is weak due to a lack of explanation.

b)      The result analysis is poor and subjective due to a lack of contributions and thorough discussion. Please rewrite it and consider providing a new discussion section to provide significant criticism and research limitations.

c)       Authors should answer your research question in the conclusions and discussion. Please provide a reasonable need to read your work’s results than previous ones or simply answer what we learned compared with current, significant research (up to 2022 should be your work’s “significance”).

d)      How general are your results and how do you believe that such findings have to be of global interest? Please relate these with your limitations and Discussion that is not exist. Why?

e)      Are there any points of view related to the consequences of this study’s limitations that may have an impact on their findings?

4.    Conclusions and limits are too short for such a study.

a)       Are there any points of view related to the consequences of this study’s limitations that may have an impact on their findings?

b)      Implications for practice and method are not provided.

-

Author Response

Dear esteemed reviewers,

We express our profound gratitude for your diligent and insightful assessment of our manuscript. Your valuable recommendations have greatly enhanced the rigor and coherence of our work. On behalf of the entire team, I extend my heartfelt appreciation for your dedicated effort in reviewing our paper. Textual amendments based on your constructive feedback are highlighted in yellow throughout the manuscript, signifying the incorporation of your invaluable suggestions.

Below, we provide a comprehensive response to each of your comments and suggestions:

Reviewer 2; Comments to the Author:

The novelty and contribution of this paper cannot be found inside the text; thus, some changes are needed. The research approach is very weak, and the findings need to be strengthened.

Thank you very much for your feedback. We hope our implemented changes will support your decision to help us publish our work. We have done our best to update the paper with the help of your review and the other two reviewers (see Reviewers 1 and 3). We sincerely hope that our improvements to the manuscript are taken up positively.

Here are my comments on improving the manuscript:

  1. Overall:
  2. a)Why do the authors conduct this study? The research contributions are weak. Please kindly explain.

Currently, there is a great need for the testing and evaluation of competency-based learning models in German undergraduate medical education. The Medical Faculty Association of the Federal Republic of Germany explicitly asks for the assistance of medical faculty in its newest publication of competency-based learning objectives catalog and states: “Locally anchoring the competencies may require new learning activity and examination formats. These must be carefully evaluated and checked for feasibility.” (Medizinische Fakultäten.de (mft) [online database]; https://medizinische-fakultaeten.de/themen/studium/nklm-nklz/).

Our study caters to that need. It provides a student voice to the process of implementing a national competency based undergraduate medical education curriculum.

In Canada, and other countries too, the focus is on testing various curricula models. (Dagnone, J. D., Chan, M. K., Meschino, D., Bandiera, G., Den Rooyen, C., Matlow, A., ... & St Croix, R. (2020). Living in a world of change: bridging the gap from competency-based medical education theory to practice in Canada. Academic Medicine95(11), 1643-1646.)

Ratnaplan and Hilliard have also dealt with the planning of learning activities in continuing medical education. They believe that a so-called needs-assessment is a fundamental step to ensure the relevance of learning activities to students.

With projects such as this study, it is possible to find out what the learners want and what motivates them to learn (Ratnapalan S, Hilliard RI (2002) Needs assessment in postgraduate medical education: a review. Medical Education Online 7:4542)

Publications on student experiences are extremely important in this case so that other faculties can benefit from each other's insights when redesigning their courses.

Our contribution is to report, in the sense of a needs-assessment, how students have experienced the impact of individual teaching formats and events on their motivation. We have tried to highlight this in the manuscript.

If you need further details on our work or our overall research strategy, I have added some other publications that may help with a better understanding of our local work:

https://link.springer.com/article/10.53180/zfa.2022.0396-0401

https://link.springer.com/article/10.53180/zfa.2022.0143-0147

https://link.springer.com/article/10.53180/zfa.2022.0229-0233

)      The research structure is not appropriate for a scientific article, e.g., research questions are missing from the introduction, results and conclusion as well as the discussion is not written in an “appropriate” manner. Please update.

We thank you for your advice. We have made every effort to bring all sections up to date and include research questions or make the preexisting ones more recognizable.

  1. c)Please consider how to effectively integrate some review papers and update.

Many thanks for this recommendation. We looked at various reviews during the post-processing phase and integrated them into the text. We marked them in yellow for your convience.

  1. Introduction:
  2. a)Research questions, that drive the paper, should be built in the introduction from an ongoing and pertinent bibliography (up to 2022-23) and these should be of global interest and not focused on a particular local problem. Identifying a research gap is the most important by indicating in-text some newer references that are significant to your particular field of research.

Thank you very much for your comment. The bibliography has been revised and several new sources have been added with a focus on international research fields. The new references are highlighted in yellow for your convenience. The current statements of the German NKLM were still kept into consideration. Even though national curricular reforms may have a local impetus, we strongly feel that it contributes to the international realm of medical education research that accompanies such reforms. We believe that it may therefore be interesting for other nationalities to have an idea of how CBME is academically approached in Germany.

  1. b)Due to a weak contribution, please pay attention to addressing new research gaps and then emphasize why the authors do this study.

Thank you for your comment. We clarified the research gaps and hope you feel more positive about the implemented changes. We also elucidated on the reasons for conducting this study.

  1. Discussion:
  2. a)The proposed reason for conducting such a study is weak due to a lack of explanation.

 Thank you for your comment. We have further elaborated on the explanations at your suggestion.

  1. b)The result analysis is poor and subjective due to a lack of contributions and thorough discussion. Please rewrite it and consider providing a new discussion section to provide significant criticism and research limitations.

 Thank you for your comment. We were a bit surprised by your wording. Subjectivity is in the nature of qualitative research, as the analysis of the data is always an interpretation of the researcher(s). May I quote:

“Quantitative research has a positivist paradigm, in which the world to be researched is viewed as an objective reality, but qualitative research has a naturalistic paradigm, in which the world to be researched is viewed as a socially constructed subjective reality”  -Quantitative and qualitative methods in medical education research: AMEE Guide No 90: Part I- https://doi.org/10.3109/0142159X.2014.915298

Prior to initiation this research project, we intentionally reflected on the differen research paradigms relevant to medical education and particularly to this project:

-Bergman, E., de Feijter, J., Frambach, J., Godefrooij, M., Slootweg, I., Stalmeijer, R., & van der Zwet, J. (2012). AM last page: A guide to research paradigms relevant to medical education. Academic medicine : journal of the Association of American Medical Colleges87(4), 545. https://doi.org/10.1097/ACM.0b013e31824fbc8a

 We have taken great care to adhere to the recommended quality criteria for qualitative research.

-Frambach, J. M., van der Vleuten, C. P., & Durning, S. J. (2013). AM last page. Quality criteria in qualitative and quantitative research. Academic medicine : journal of the Association of American Medical Colleges, 88(4), 552. https://doi.org/10.1097/ACM.0b013e31828abf7f

We carried out investigator triangulation sessions (several rounds), member checking and the preformed codes were revaluated under the consideration of the theoretical framework of motivation in two interdisciplinary qualitative research sessions at the institute of FM (UdS).The methods & discussion section has been updated to make this process and also its limitations clearer.

  1. c)Authors should answer your research question in the conclusions and discussion. Please provide a reasonable need to read your work’s results than previous ones or simply answer what we learned compared with current, significant research (up to 2022 should be your work’s “significance”).

The discussion section has been updated by adding comparisons with findings from other studies and new conclusions resulting from our research. We hope you feel that we have reasonably adhered to your feedback.

  1. d)How general are your results and how do you believe that such findings have to be of global interest? Please relate these with your limitations and Discussion that is not exist. Why?

      Thank you very much for your comment. We are aware and we have tried to make this clear in the text, that these are only possible effects on motivation and certainly not universally valid effects (as is common for qualitative work). The aim of qualitative studies is usually not to generate generalizable data, but to highlight possible effects that can be taken into account in further quantitative studies or in new implementations, or that can be explored further.

(see also: constructivism“Knowledge is gained by an inductive approach: recognizing, understanding, developing, and contrasting constructions through dialogue.”)

-Frambach, J. M., van der Vleuten, C. P., & Durning, S. J. (2013). AM last page. Quality criteria in qualitative and quantitative research. Academic medicine : journal of the Association of American Medical Colleges, 88(4), 552. https://doi.org/10.1097/ACM.0b013e31828abf7f

  1. e)Are there any points of view related to the consequences of this study’s limitations that may have an impact on their findings?

Yes, thank you for this question.  E.g., Covid-19 may have had an impact on students' views. This limitation and its possible consequences, as well as other limitations were elucidated on and explained in more detail.

  1. Conclusions and limits are too short for such a study.

Please be aware that based on the journal’s publication guidelines, a succinct and focused conclusion is a key element of their publication criteria.

  1. Are there any points of view related to the consequences of this study’s limitations that may have an impact on their findings?

Thank you for this question. We assume that you may have accidentally copy’/pasted this point twice. You can find the answer to this comment above.

  1. Implications for practice and method are not provided.

The following implications can be found in the discussion section: for your convenience, we have copied them into this response letter.

- This study reveals three important aspects that ought to be considered if educators aim to purposefully foster students’ motivation. These are: The role of instructional design in motivation, the role of emotional motivation transfer between staff and students and the role of perceived professional growth while learning.

- A blended learning concept, which combines online learning and face-to face learning, appears to meet the idetified needs of students in a CBME background

- the choice of learning media seemed to play less of a role for motivation than the content and its visual appeal

- a certain amount of variety through podcasts, videos and learning cards was seen as positive, on the other hand it seems to have a negative effect on self-efficacy if the materials provided are seen as too time-consuming to process. Teachers planning the implementation of a competency-based curriculum and seeking to maximize media diversity to appeal to as many students as possible should be aware of this possible negative effect.

- We agree with Ryan et al. that teachers should try not to overload students with excessive content. However, we have found that it seems to be possible to emphasize clinical relevance through simple methods, such as linking topics to relevant clinical cases

- that simulation seminars seem to be a good fit for CBME for reasons of possible enhancement of self-efficacy and its effectiveness in teaching important relatedness promoting practical skills such as interpersonal interaction and empathy.

[final notes]

Dear reviewers,

We would like to express our sincere gratitude for your valuable feedback and support in reviewing our paper. Your constructive comments and suggestions have significantly enhanced the quality and clarity of our work. We appreciate the time and effort you have invested in providing such insightful reviews.

We have carefully considered all your comments and made the necessary revisions to address the issues raised. In response to Reviewer 1's question regarding the impact of the COVID-19 pandemic, we have added a detailed explanation of how the pandemic influenced our curriculum design and data collection, and how it may have shaped participants' views on relational learning.

Additionally, we have rephrased certain sections to better convey our research intentions and contributions. We have integrated relevant review papers and updated our bibliography to reflect the most recent and pertinent literature in the field of medical education.

In the Methods section, we have provided a clearer explanation of the ASSIST item questions and their connection to the RASI instrument to avoid confusion for readers.

Moreover, we have expanded the discussion section to address the limitations of our study and the implications for practice and research. We have also added a more comprehensive conclusion that summarizes the main findings and their significance in the context of current research.

Regarding the term "blended learning," we have included a well-defined explanation with proper references to ensure a clear understanding of its meaning in our study.

We have also taken the opportunity to clarify points raised by Reviewer 3, including providing additional context for certain statements and refining Table 3 for better comprehension.

Once again, we extend our heartfelt appreciation for your academic input and support of our medical education research. Your contributions have been instrumental in refining our work, and we hope that the improved manuscript now meets the standards for publication in your esteemed journal.

Thank you for your valuable guidance, and we look forward to the possibility of sharing our findings with the wider medical education community.

Sincerely

On behalf of the entire authors team

XXX

Reviewer 3 Report

The topic of what drives motivation in medical students is interesting and worthy of study. This is a small study involving less than 20 students at a single institution, so it is far from generalizable. However, the authors note these limitations and report their findings within the frame of self-determination theory. The literature is well synthesized, the paper is well written, and the authors do not overstate their results. 

There are a few specific issues that should be addressed:

The study took place in the middle of the pandemic (2020-21): How did this influence the curriculum and student contact with each other and faculty?

The authors use the term "blended learning" on several occasions - this term needs to be carefully defined as it can mean different things to different people.

Page 4/line 133 (Methods) - The use of the ASSIST item questions needs to be described. The authors begin by describing the RASI instrument and then ASSIST is mentioned without any context or explanation. The reviewer had to consult reference 20 to decipher this part of the methods section.

Page 7, line 240 - what is the evidence that negative feedback led to externalization of motivation in this student?

Table 3, B3 states "and for me, this led..." What is "this"?

Overall the English is fine, just one comment:

Table 2, middle column - "getting enforced to reflect..." - Not sure what this means. Forcing students to reflect? (that seems rather strong). Perhaps  "Promoting reflection on a subject"?

Author Response

Dear esteemed reviewers,

We express our profound gratitude for your diligent and insightful assessment of our manuscript. Your valuable recommendations have greatly enhanced the rigor and coherence of our work. On behalf of the entire team, I extend my heartfelt appreciation for your dedicated effort in reviewing our paper. Textual amendments based on your constructive feedback are highlighted in yellow throughout the manuscript, signifying the incorporation of your invaluable suggestions.

Below, we provide a comprehensive response to each of your comments and suggestions:

Reviewer 3; Comments to the Author:

The topic of what drives motivation in medical students is interesting and worthy of study. This is a small study involving less than 20 students at a single institution, so it is far from generalizable. However, the authors note these limitations and report their findings within the frame of self-determination theory. The literature is well synthesized, the paper is well written, and the authors do not overstate their results. 

Thank you for your appreciative comment. We are pleased that you like our work. It is ever so important to provide constructive or positive feedback, during this very long and complex process of a research project. Our team is highly grateful for your work and your enthusiasm for this work. Thank you.

There are a few specific issues that should be addressed:

The study took place in the middle of the pandemic (2020-21): How did this influence the curriculum and student contact with each other and faculty?

Thank you very much for this highly relevant element. Indeed, this is a shortcoming we had to address. Reviewer 1 has already correctly pointed this out to us and the text has been completed in the introduction and discussion.

The authors use the term "blended learning" on several occasions - this term needs to be carefully defined as it can mean different things to different people.

Very true, especially during the last couple of years the use of this term has become somewhat diluted. We added the definition of blended learning with references to the introduction.

Page 4/line 133 (Methods) - The use of the ASSIST item questions needs to be described. The authors begin by describing the RASI instrument and then ASSIST is mentioned without any context or explanation. The reviewer had to consult reference 20 to decipher this part of the methods section.

This is highly important, thank you. We have now added an explanation which we hope makes it clear that the RASI is part of the ASSIST questionnaire.

Page 7, line 240 - what is the evidence that negative feedback led to externalization of motivation in this student?

Thank you very much, in fact the students describe a decrease in self-efficacy as an example quote:

B12: ‘(…) our performance wasn’t even that bad, but the feedback was purely negative. That’s what I found a bit annoying (…).’ (Interview 2, transcript, pos. 222)

 [in relation to the received /perceived negative feedback during a simulation-- One participant reacted defensively to receiving only negative feedback.

One participant responded defensively upon receiving solely negative feedback. Despite not seeking to rationalize the evaluation of her performance, there was a noticeable decline in her belief in her own capabilities (perceived self-efficacy). This observation suggests that an exclusive focus on negative feedback may impede the development of perceived self-efficacy, consequently affecting intrinsic motivation.]".

 A lower level of self-efficacy is associated with a decrease in intrinsic motivation. We made sure this connection and our interpretation thereof is more clearly explained in the text. (marked in yellow)

Table 3, B3 states "and for me, this led..." What is "this"?

Thank you very much. You are right, this statement needs to be described in more detail. B3 uses "this" to refer to her previous statement that she found the course preparation time-consuming. We have added this essential element to Table 3.

[final notes]

Dear reviewers,

We would like to express our sincere gratitude for your valuable feedback and support in reviewing our paper. Your constructive comments and suggestions have significantly enhanced the quality and clarity of our work. We appreciate the time and effort you have invested in providing such insightful reviews.

We have carefully considered all your comments and made the necessary revisions to address the issues raised. In response to Reviewer 1's question regarding the impact of the COVID-19 pandemic, we have added a detailed explanation of how the pandemic influenced our curriculum design and data collection, and how it may have shaped participants' views on relational learning.

Additionally, we have rephrased certain sections to better convey our research intentions and contributions. We have integrated relevant review papers and updated our bibliography to reflect the most recent and pertinent literature in the field of medical education.

In the Methods section, we have provided a clearer explanation of the ASSIST item questions and their connection to the RASI instrument to avoid confusion for readers.

Moreover, we have expanded the discussion section to address the limitations of our study and the implications for practice and research. We have also added a more comprehensive conclusion that summarizes the main findings and their significance in the context of current research.

Regarding the term "blended learning," we have included a well-defined explanation with proper references to ensure a clear understanding of its meaning in our study.

We have also taken the opportunity to clarify points raised by Reviewer 3, including providing additional context for certain statements and refining Table 3 for better comprehension.

Once again, we extend our heartfelt appreciation for your academic input and support of our medical education research. Your contributions have been instrumental in refining our work, and we hope that the improved manuscript now meets the standards for publication in your esteemed journal.

Thank you for your valuable guidance, and we look forward to the possibility of sharing our findings with the wider medical education community.

Sincerely

On behalf of the entire authors team

XXX

Round 2

Reviewer 2 Report

No further comments 

No further comments